# Isolation, Identification and Chemical Modification of Bufadienolides from *Bufo melanostictus* Schneider and Their Cytotoxic Activities against Prostate Cancer Cells

**DOI:** 10.3390/molecules29071571

**Published:** 2024-03-31

**Authors:** Qingmei Ye, Rong Lin, Zeping Chen, Juan Li, Caijuan Zheng

**Affiliations:** 1Key Laboratory of Tropical Medicinal Resource Chemistry of Ministry of Education, Key Laboratory of Tropical Medicinal Plant Chemistry of Hainan Province, College of Chemistry and Chemical Engineering, Hainan Normal University, Haikou 571158, China; qingmei-ye@hainmc.edu.cn; 2Hainan General Hospital & Hainan Affiliated Hospital of Hainan Medical University, Haikou 570311, China; 3Hubei Province Key Laboratory of Traditional Chinese Medicine Resource and Chemistry, Department of Pharmacy, Hubei University of Chinese Medicine, Wuhan 430065, China; candis1005@163.com; 4Institute of Traditional Chinese Medicine and Natural Products, College of Pharmacy, Jinan University, Guangzhou 510632, China; 15602275702@163.com

**Keywords:** toad venom, bufadienolides, cytotoxic activity

## Abstract

The traditional Chinese medicine toad venom (*Venenum bufonis*) has been extensively used to treat various diseases, including cancers, in China and other Southeast Asian countries. The major constituents of toad venom, e.g., bufadienolides and alkaloids, exhibit broad-spectrum pharmacological effects in cancers. Herein, two new bufadienolides (**1** and **2**), along with eleven known compounds (**3**–**13**) were successfully isolated from *Bufo melanostictus* Schneider. Their structures were elucidated by extensive spectroscopic data and X-ray diffraction analysis. Furthermore, four lactam derivatives were synthesized through the transformation of bufadienolides lactones. The inhibitory effects of these compounds against human prostate cancer cell lines PC-3 and DU145 were evaluated. The outcomes indicated a notable trend, with a substantial subset displaying nanomolar range IC_50_ values against PC-3 and DU145 cells, underscoring their pronounced cytotoxicity. Moreover, a noteworthy distinction surfaces, wherein lactones consistently outperformed their lactam counterparts, further validating their heightened potency for the treatment of prostate cancer. This study contributes significant preclinical evidence substantiating the therapeutic viability of bufadienolides and toad venom as intervention strategies for prostate cancer.

## 1. Introduction

Toad venom, known as *Venenum bufonis*, has been utilized for therapeutic purposes for many centuries in modern China as well as other Asian nations including South Korea and Japan, following it being appropriately dried and formulated [1,2]. Toad venom plays a central role as the primary constituent in various therapeutic formulations or Chinese patent medicines [3]. Of note, the cinobufacini injection, as a sanctioned therapy, has been used in the management of cancer, including lung and liver malignancies, in China [4,5]. Research has elucidated that the pharmacologically active components within toad venom chiefly comprise alkaloids and bufadienolides, which are distinguished for their diverse bioactive activity for different diseases [6,7,8,9]. Previous studies have shown that bufadienolides demonstrate potent cytotoxic activities against diverse cancer cell lines in vitro and in vivo, with IC_50_ values in the nanomolar range [10]. Additionally, many studies have revealed some of the structure–activity relationship of bufadienolides in toad venom in cancer cells. One study showed that the 14*β*-hydroxy derivatives are more active than the 14,15-epoxy compounds, which is the fundamental bufadienolide skeleton structure that is critical for maintaining its cytotoxic activity, while the 3-hydroxyl epimerization leads to a significant decrease in terms of toxicity [11]. While a significant number of bufadienolides have been identified from toad venom derived from different origins, their chemical profiles remain incompletely elucidated, and thus more studies are needed to reveal all necessary information for wider clinical use [12,13,14]. Previous studies have shown that the dichloromethane layer of toad venom has very strong activity against PC-3 and DU145 cells with IC_50s_ < 0.011 μM [15], which may suggest that the organic layers are the main bioactive components to be isolated and characterized. In the current study, two new bufadienolides (**1** and **2**), along with 11 analogues (**3**–**13**) were isolated from the *B. melanostictus* Schneider by the bioassay-guided phytochemical method (Figure 1). Notably, four lactam derivatives (**14**–**17**) were semi-synthesized from the corresponding lactone structures (**3**, **5**, **7** and **13**) to augment the stability of these lactones. All compounds have been evaluated for their cytotoxic activity against human prostate cancer cell lines PC-3 and DU145. Herein, the isolation, structural elucidation, and cytotoxic activity of these compounds are described.

## 2. Results and Discussion

### 2.1. Structural Elucidation of the Isolated Compounds

Compound **1** was obtained as a white powder. The molecular formula C_26_H_36_O_6_ of **1** was determined by its HR-ESI-MS data [*m*/*z* 467.2319 [M + Na]^+^, calculated for 467.2410]. The ^1^H NMR data of **1** exhibited three olefinic protons at *δ*_H_ 7.99 (1H, dd, *J* = 12.0, 4.0 Hz), 7.43 (1H, d, *J* = 4.0 Hz), and 6.28 (1H, d, *J* = 12.0 Hz), one oxygenated proton signal at *δ*_H_ 4.06 (1H, s); one methylene group at *δ*_H_ 3.88 (1H, d, *J* = 12.0 Hz) and 3.37 (1H, d, *J* = 12.0 Hz)]; one methyl group at *δ*_H_ 0.71 (3H, s) (see details in Table 1). The ^13^C NMR and DEPT-135 data of **1** showed the presence of 26 carbon resonances, including one carbonyl carbon at *δ*c 164.8, four olefinic carbons at *δ*_C_ 150.5, 149.4, 125.0, and 115.4, ten methylene carbons at *δ*_C_ 65.4, 42.2, 33.0, 31.4, 30.3, 27.2, 25.7, 24.9, 22.6 and 22.1, five methine carbons at *δ*_C_ 72.1, 52.3, 42.8, 36.6 and 30.3, and two methyl carbons at *δ*_C_ 21.3 and 17.3. The comparison of the ^1^H and ^13^C NMR of **1** with the known compound 19-hydroxybufalin [16] suggested that their structures are very similar, except for the presence of one carbonyl signal at *δ*c 173.3 and methyl signal at *δ*c 21.3 in **1**.

In the ^1^H-^1^H COSY (Correlated Spectroscopy) spectrum of **1**, three spin systems [H-1 (*δ*_H_ 1.82)↔ H-2 (*δ*_H_ 1.68) ↔ H-3 (*δ*_H_ 5.05) ↔ H-4 (*δ*_H_ 2.00) ↔ H-5 (*δ*_H_ 2.16) ↔ H-6 (*δ*_H_ 1.84) ↔ H-7 (*δ*_H_ 1.87) ↔ H-8 (*δ*_H_ 1.68)↔ H-9 (*δ*_H_ 1.80) ↔ H-11 (*δ*_H_ 1.22)↔ H-12 (*δ*_H_ 1.50), H-15 (*δ*_H_ 1.73) ↔ H-16 (*δ*_H_ 2.18) ↔ H-17 (*δ*_H_ 2.56) and H-22 (*δ*_H_ 7.99) ↔ H-23 (*δ*_H_ 6.28)] were observed. In the HMBC (Heteronuclear Multiple Bond Correlation) spectrum, the key correlations from H-26 (*δ*_H_ 2.04) to C-25 (*δ*c 172.7), from H-3 (*δ*_H_ 5.05) to C-25 (*δ*c 172.7) suggested that an acetyl group linked at the C-3 position of **1** (Figure 2). The relative configuration of **1** was established by the NOESY data. In the NOESY (Nuclear Overhauser Effect Spectroscopy) spectrum of **1**, the correlations of H-19 (*δ*_H_ 3.89) and H-8 (*δ*_H_ 1.80), H-8 (*δ*_H_ 1.80) and H_3_-18 (*δ*_H_ 0.71), H_3_-18 (*δ*_H_ 0.71) and H-21 (*δ*_H_ 7.43) indicated that these H-18, H-19 and H-21 are placed at *β* configurations. The correaltions of H-3 (*δ*_H_ 5.05) and H-4 (*δ*_H_ 2.00), H-4 (*δ*_H_ 1.48) and H-9 (*δ*_H_ 1.80)/H-7 (*δ*_H_ 1.22), H-12*α* (*δ*_H_ 1.50) and H-15 (*δ*_H_ 2.16)/H-17 (*δ*_H_ 2.56) indicated that these positions are at *α* configurations (Figure 3). To support the configurations of **1**, an X-ray crystal structure with a Flack parameter of 0.27(13) was obtained (see Figure 4). Hence, the structure of **1** was determined and named as 3-acetylbufalin.

Compound **2** was isolated also as a white powder, with the molecular formula of C_26_H_36_O_6_ based on the HR-ESI-MS spectrum (*m/z* 467.2319 [M + Na]^+^, calculated for C_26_H_36_O_6_Na: 467.2410). The ^1^H and ^13^C NMR spectra of 2 were similar to those of 1, except for the significant chemical shifts at C-3 and C-19, indicating the acetyl group of 2 was located at a different position. In the ^1^H-^1^H COSY spectrum of **2**, three spin systems were observed and drawn in boldface as detailed in Figure 2. In the HMBC spectrum, the key correlations from H-26 (*δ*_H_ 2.07) to C-25 (*δ*c 173.3), from H-19 (*δ*_H_ 4.38) to C-25 (*δ*c 173.3) suggested that an acetyl group located at the C-19 in **2**. In the NOESY spectrum of **2**, the correlations of H-19 (*δ*_H_ 4.38) and H-8 (*δ*_H_ 1.68), H-8 (*δ*_H_ 1.68), and H-18 (*δ*_H_ 0.72) suggested they were located at *β* configurations. Furthermore, the correlations of H-3 (*δ*_H_ 4.06) and H-4 (*δ*_H_ 1.98), H-4 (*δ*_H_ 1.98) and H-9 (*δ*_H_ 1.87)/H-7 (*δ*_H_ 1.65), H-12*α* (*δ*_H_ 1.53) and H-14 (*δ*_H_ 2.16)/H-17 (*δ*_H_ 2.56) indicated that these positions are at *α* configurations. Thus, the structure of **2** was elucidated to be 19-acetoxybufalin.

By comparing the physical and spectroscopic data with literature values, 11 known compounds were identified as 19-hydroxybufalin (**3**) [16], arenobufagin (**4**) [17], bufalin (**5**) [18], hellebrigenin (**6**) [19], resibufogenin (**7**) [20], bufotalin (**8**), marinobufagin (**9**) [21], cinobufotalin (**10**) [17], telocinobufagin (**11**) [22], gamabufotalin (**12**) [23], and cinobufagin (**13**) [24] as shown in Figure 1.

### 2.2. Spectroscopic Data

3-Acetylbufalin (**1**): ${\left[\alpha \right]}_{D}^{25}$-0.01 (*c* 0.1, CH_3_OH); IR (KBr) *ν*_max_ 3416 (OH), 1735 (C=O) cm^−1^; ^1^H NMR (400 MHz, MeOD) and ^13^C NMR (100 MHz, MeOD), see Table 1; HR-ESI-MS *m/z* 467.2391 [M + Na]^+^ (calcd for C_26_H_36_O_6_Na, 467.2410).

19-Acetoxybufalin (**2**): ${\left[\alpha \right]}_{D}^{25}$-0.04 (*c* 0.1, CH_3_OH); IR (KBr) *ν*_max_ 3420 (OH), 1735 (C=O) cm^−1^; ^1^H NMR (400 MHz, MeOD) and ^13^C NMR (100 MHz, MeOD), see Table 1; HR-ESI-MS *m*/*z* 467.2319 [M + Na]^+^ (calcd for C_26_H_36_O_6_Na, 467.2410).

### 2.3. X-ray Analysis

Crystal data for **1**: C_26_H_36_O_6_, *M_r_* = 444.55, triclinic, *a* = 8.5025(5) Å, *b* = 8.5651(5) Å, *c* = 8.7725(5) Å, *α* = 72.201(5)°, *β* = 77.341(5)°, *γ* = 77.229(5)°, *V* = 585.27(6) Å^3^, space group *P* 1, Z = 1, *D_x_* = 1.261 mg/mm^3^, *μ* (Cu K*α*) = 0.715 mm^−1^, and F(000) = 240. Independent reflections: 3917 (*R*_int_ = 0.0339). The final *R*_1_ values were 0.0358, *wR*_2_ = 0.0954 (*I* > 2*σ*(*I*)). Flack parameter = 0.27(13). CCDC number 2288954.

### 2.4. Synthesis of Lactam Derivatives of BDs

It is known that two prominent limitations warrant consideration in the realm of bufadienolides regarding their developability or drug-likeness: the inherent toxic effects and physicochemical stability concerns arising majorly from the lactone moiety which is easily and readily undergoing hydrolysis procedure in mild acidic or basic condition or in physiological condition. In light of these challenges, a strategic medicinal–chemistry approach was undertaken to address these issues. Notably, the transformation of bufadienolides with a lactone moiety into counterparts bearing a lactam structure was pursued.

The rationale underpinning this approach stems from the recognized superior stability and prevalence of the amide bond (CONH) in a lactam moiety in contrast to the ester bond (OCO) in lactone presented in bufadienolides. This transformation engenders the potential to ameliorate the inherent toxic effects and the stability quandary associated with the lactone motif.

To this end, four lactam derivatives, 19-hydroxybufalin-lactam (**14**), bufalin-lactam (**15**), resibufogenin-lactam (**16**), and cinobufagin-lactam (**17**) were successfully synthesized via a one-step chemical reaction (see Figure 5A for chemical condition and Figure 5B for the structures of synthesized lactams). Four lactam derivatives were evaluated for their cytotoxicity against human prostate cancer PC-3 and DU145 cell lines.

### 2.5. The Anti-Proliferative Activities of All Isolated Compounds ***1***–***13*** and Four Modified Compound in Prostate Cancer Cell Lines

The cytotoxic activity of the isolated compounds (**1**–**13**) and four modified compounds (**14**–**17**) were evaluated for their cytotoxic activity against human prostate cancer cell lines PC-3 and DU145 using MTT assay. The results in Table 2 showed that nearly all isolated compounds had potent cytotoxic effects against PC-3 and DU145 cells with IC_50_ values < 0.5 μM. Among them, compounds **3**, **5** and **12** showed the strongest activity against these two cancer cells with IC_50_ values < 0.02 μM. However, the two new compounds **1** and **2** showed lower cytotoxicity than Paclitaxel in both cell lines. Notably, lactone-bearing compounds demonstrated markedly heightened activity against prostate cancer cells in contrast to their lactam counterparts **14**–**17** (see details in Table 2).

It is also worth noting that several bufadienolides demonstrated higher (**3**, **5**, and **12,** 2–4 folds) or comparable (**10**) potentials in PC-3 cells compared to the positive control Paclitaxel, but all four showed lower cytotoxicity in DU145 cells than Paclitaxel. Interestingly, both PC-3 and DU145 cells are androgen receptor-independent cell lines that can both be categorized as castration-resistant prostate cancer cells. It is thus unknown why these compounds showed different cytotoxic profiles toward these two similar cell lines. Further studies are needed to decipher the underlying mechanisms. These results may suggest that the cytotoxicity of bufadienolides is cell type-dependent and that compounds **3**, **5**, and **12** can be further evaluated in PC-3 cells xenograft mice models.

We have used the human nonmalignant prostate epithelial RWPE-1 cell line for parallel comparison. All compounds showed similar toxicity in RWPE-1 cells compared to the two prostate cancer cell lines. However, when we used the African green monkey kidney-derived Vero cell line, the compounds showed much lower toxicity than in prostate cancer cell lines. More work is needed to address the toxic issues.

### 2.6. The Pharmacophore Modeling of BDs against Prostate Cancer Cells

Subsequently, the structure–activity relationship of all compounds against two prostate cancer cells was discussed using molecular dynamics simulation by computer-aided drug design software. This analytical approach aimed to unravel the intricate interplay between the compounds’ structural attributes and their activity profiles which may provide essential information for further structural modification.

The outcomes of this endeavor disclosed that bufadienolides harbor two distinctive categories of features, namely hydrophobic groups and hydrogen bond donors as shown in Figure 6. Notably, the positioning of hydrogen bond receptors has emerged as a pivotal determinant in maintaining the cytotoxic efficacy of bufadienolides. Structurally, changes in the cyclopentane conformation affect the size of the molecule, thereby influencing the overlap with hydrophobic and hydrogen bond feature elements, a revelation corroborated by the insights gleaned from Figure 6. Compared to bufalin (**5**), bufotalin (**8**) (16*β*-acetoxyl) exhibited decreased activity, which contrasts with previous reports [14] and may be attributed to the variety of cancer cells used in the study. The transformation of the bufadienolide structure (**5**) into an internal amide (**15**) resulted in a significant decrease in activity. The disruption of the lactone structure led to a decrease in activity, highlighting the crucial role of the lactone structure in the activity of these compounds, and that it should be intact while undergoing further medicinal–chemistry study.

## 3. Methods and Materials

### 3.1. General Experimental Procedures

The optical rotations were measured on a JASCO P-1020 digital polarimeter (JASCO, Tokyo, Japan) using a thermostable optical glass cell (0.1 dm path length and c in g/100 mL). Melting points were determined on a WRX-4 micro melting point apparatus (Shanghai YiCe Apparatus and Equipment Co., Ltd., Shanghai, China) and were uncorrected prior to use. Ultraviolet (UV) spectra were recorded on a U-3900 UV-VIS spectrophotometer (Hitachi, Ltd., Tokyo, Japan). The infrared (IR) spectra were recorded on a Thermo Scientific Nicolet 6700 using the KBr disks spectrophotometer (Thermo Scientific, Madison, WI, USA). Single-crystal data were measured by an Agilent Gemini Ultra X-ray single-crystal diffractometer (Cu Kα radiation). 1D and 2D NMR spectra were recorded from a Bruker AV spectrometer (400 MHZ for ^1^H and 100 MHZ for ^13^C) and a JNM-ECZS spectrometer (600 HMZ for ^1^H and 150 MHZ for ^13^C). HR-ESI-MS spectra were obtained from a Q-TOF Ultima Global GAA076 LC mass spectrometer (Waters Corporation, Milford, MA, USA). Preparative high-performance liquid chromatography (HPLC) of Agilent 1100 prep-HPLC system with an Agilent C_18_ analytical HPLC column was used for purification (Agilent Technologies, Waldbronn, Germany). Sephadex LH-20 (Pharmacia Co. Ltd., Sandwich, UK) and silica gel (200−300 and 300−400 mesh, Qingdao Marine Chemical Factory, Qingdao, China) were used for column chromatography (CC). All chemical solvents (analytic grade) were purchased from Xilong Chemical Reagent Factory (Guangzhou, China).

### 3.2. Materials and Reagents

The toad venom of *Bufo melanostictus* Schneider was purchased from Guilin of Guangxi province, in February 2020. The employed materials and reagents encompassed 3-[4,5-dimethylthiazol-2-yl]-2,5 diphenyl tetrazolium bromide (MTT, Sigma, USA, Lot No. MKBL6647V), dimethyl sulfoxide (DMSO, Sigma, MO, USA, Lot No. WXBB3106V), Ham’s F-12 Basal Medium (Procell, Wuhan, China, Lot No. WH1022E131), MEM (inclusive of NEAA) basal medium (Procell, Wuhan, China, Lot No. WH0023X131), fetal bovine serum (Four Seasons, Hangzhou, China, Lot No. 22090701), Trypsin (Gibco, NY, USA, batch No. 2428760), sodium chloride (Sinopharm Chemical Reagent Co. Ltd., Shanghai, China, Lot No. 20161025), and disodium hydrogen phosphate anhydrous (Sinopharm Chemical Reagent Co., Ltd., Shanghai, China, Lot No. 20210301). All mentioned substances were procured from commercially established suppliers.

### 3.3. Extraction and Isolation

The dry toad venom (1.0 kg) was extracted by 95% ethanol under ultrasound conditions under room temperature. After filtration, the resulting product was further concentrated under reduced pressure to yield the crude extract (260 g). Additionally, the crude extract was dissolved in 10% MeOH–H_2_O and subsequently partitioned by CH_2_Cl_2_ to obtain the organic layer (90.6 g) according to a previous study [15]. The CH_2_Cl_2_ layer was subjected to a silica gel column with a gradient mixture of petroleum ether and ethyl acetate to finally yield 15 fractions. Fraction 4 was separated over an ODS column and further purified by the preparative HPLC with MeOH and water to yield compounds **8** (14.8 mg), **9** (16.0 mg), and **13** (41.0 mg). Similarly, compounds **4** (9.6 mg), **5** (40.5 mg), **7** (39.4 mg), **10** (17.7 mg), **11** (18.0 mg), and **12** (16.3 mg) were obtained by using a reverse C_18_ column and preparative HPLC with MeOH and water in fraction 6. Additionally, fraction 7 was separated by preparative HPLC to produce compounds **1** (3.2 mg) and **2** (4.1 mg). Fraction 10 was separated by preparative HPLC using MeOH and water to obtain compounds **3** (48.9 mg) and **6** (11.2 mg).

### 3.4. X-ray Crystallographic Analysis of Compound ***1***

Colorless crystals of **1** were obtained from the MeOH solution. Single-crystal X-ray diffraction data were collected on an Xcalibur, Atlas, Geminiultra diffractometer with Cu K*α* radiation (*λ* = 1.54184 Å) at 293 (2) K. Crystallographic data of **1** has been deposited in the Cambridge Crystallographic Data Centre with the deposition number. Copies of the data can be obtained, free of charge, on application to the Director, CCDC, 12 Union Road, Cambridge CB2 1EZ, UK [Fax: +44-(0)1223-336033, or e-mail: deposit@ccdc.cam.ac.uk].

### 3.5. The Lactone to Lactam Conversion of the Bufadienolides

The conversion of four bufadienolides (**3**, **5**, **7**, and **13**) into their respective lactams (**14**–**17**) were accomplished by refluxing a mixture of bufadienolides and ammonium acetate (at a ratio of 3–8 equivalents) in N,N-dimethylformamide (DMF) at temperatures ranging from 100–160 °C for a duration of 0.5–3 h. Subsequent to the reaction, the target lactam was isolated by the preparative HPLC, using a mobile phase composed of MeOH and H_2_O. Comprehensive details regarding this process are documented in the Appendix A.

### 3.6. Cell Culture

Human prostate cancer PC-3 and DU145 cells were purchased from Shanghai Zhongqiao Xinzhou Biotechnology Co. (Shanghai, China). The cultivation of human prostate cancer PC-3 cells entailed their placement within Ham’s F-12 medium, supplemented with 10% fetal bovine serum and 1% penicillin-streptomycin. Similarly, human prostate cancer DU145 cells were cultured in MEM medium, supplemented with 10% fetal bovine serum and 1% penicillin-streptomycin. Both cell types were cultured within a 5% CO_2_, 37 °C incubator, under precise conditions maintained by a Carbon Dioxide Incubator (Thermo, USA) [25]. Routine medium replacement occurred every 2 days. Upon achieving approximately 80% confluence, the cells underwent a 3 min trypsinization process, and their morphological changes were monitored microscopically. The trypsinization process was halted by the introduction of a culture medium containing 10% fetal bovine serum, followed by transfer to a centrifuge tube. Subsequent centrifugation at 1000 rpm for 3 min, and then they were transferred to a new culture flask, which completed the process. For experimental purposes, cells in the logarithmic growth phase were selected.

### 3.7. Cytotoxicity Assay

Cytotoxic activities of all compounds against PC-3 and DU145 cell lines were evaluated by the MTT method. In short, PC-3 and DU145 (3000–5000 cells/well) were seeded in the 96-well plates. After overnight attachment, fresh medium with gradient concentrations of Paclitaxel as positive control and all compounds were added to the wells and cocultured for another 72 h. The cells treated with 0.1% DMSO in medium were used as the negative control, consistent with the solvent used for dissolving the sample. Then 100 μL MTT solution (0.5 mg/mL) was added to each well. At the end of treatment, the medium in each well was carefully removed, 100 μL DMSO was added into each well, and the plates were shaken for 30 s for a complete dissolve. The absorbance was measured at 490 and 570 nm using a microplate reader (Bio-Tek Instruments Inc., Winooski, VT, USA). The cell survival rate was computed using the formula: Cell survival rate (%) = (Absorbance of Experimental group/Absorbance of Control group) × 100%. The half-maximal inhibitory concentration (IC_50_) of all bufadienolides and four lactam derivatives against PC-3 cells and DU145 cells was subsequently calculated and analyzed.

### 3.8. Pharmacophore Modeling

Pharmacophore modeling was conducted employing the Common Feature Pharmacophore application within Discovery Studio 3.0, following established protocols as detailed in prior literature [26,27]. The conversion of 2D chemical structures into 3D conformations was carried out using the Conformations application, and the subsequent generation of pharmacophores was accomplished via the HipHop module. The generated pharmacophore hypotheses were capped at a maximum of 10. Feature options were defined with a minimum of 4 and a maximum of 6. Principal values were assigned as 0 (IC_50_ > 1 μM), 1 (1 μM > IC_50_ > 0.3 μM), and 2 (IC_50_ > 0.3 μM), corresponding to their cytotoxic activity levels [26,27].

## 4. Conclusions

Thirteen bufadienolides, including two new compounds **1**–**2**, were isolated from traditional Chinese medicine toad venom. In addition, four lactam derivatives of bufadienolides were obtained by chemical modification. Cytotoxicity of all compounds against prostate cancer cells PC-3 and DU145 were evaluated by the MTT method. Compounds **3**, **5** and **12** displayed very strong inhibitory effect against PC-3 and DU145 with IC_50S_ < 0.02 μM, which were lower than positive control Paclitaxel. The structure–activity relationship showed that the lactone ring of bufadienolides is important in terms of antitumor activity since the lactam derivatives of bufadienolides showed significantly decreased activities or lost activities. This study provided possible new potential anti-tumor molecules for clinical treatment for prostate cancers.

## Figures and Tables

**Figure 1 molecules-29-01571-f001:**
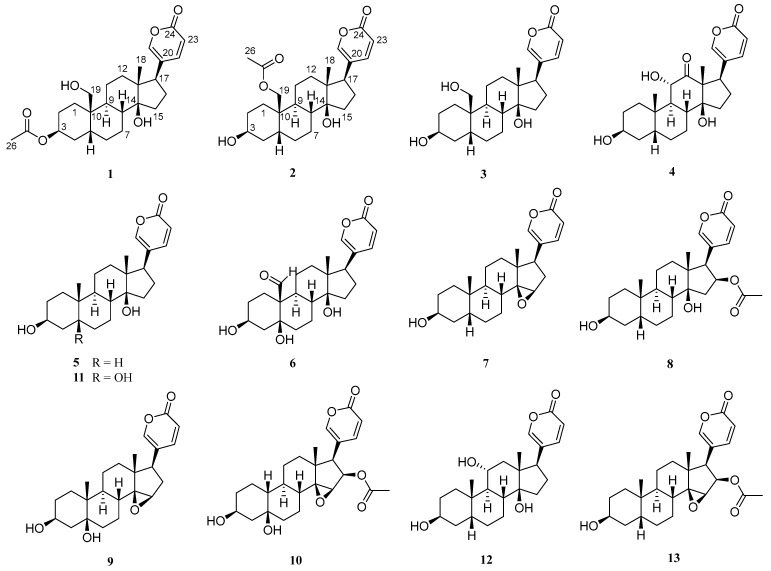
The chemical structures of compounds **1**–**13.** Compounds **1**–**2** were unreported bufadienolides, and compounds **3**–**11** were described bufadienolides.

**Figure 2 molecules-29-01571-f002:**
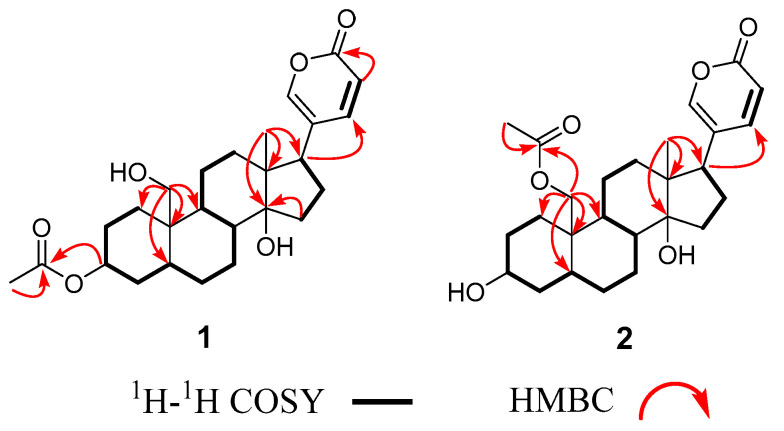
^1^H–^1^H COSY and key HMBC correlations of compounds **1** and **2**. Bold lines indicate the ^1^H-^1^H coupling, and arrows indicated ^1^H/^13^C long-range correlations.

**Figure 3 molecules-29-01571-f003:**
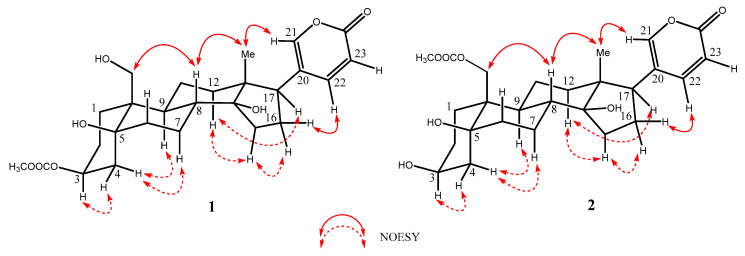
The key NOESY correlations of compounds **1** and **2**. Solid arrows indicate correlations in the *β*-orientation, and dashed arrows show correlations in the *α*-orientation.

**Figure 4 molecules-29-01571-f004:**
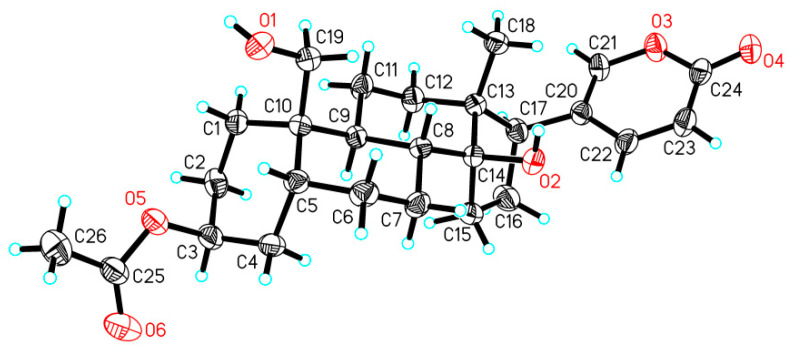
X-ray structure of **1**. Hydrogen atoms were shown as blue spheres of arbitrary size, and oxygen atoms were shown in red.

**Figure 5 molecules-29-01571-f005:**
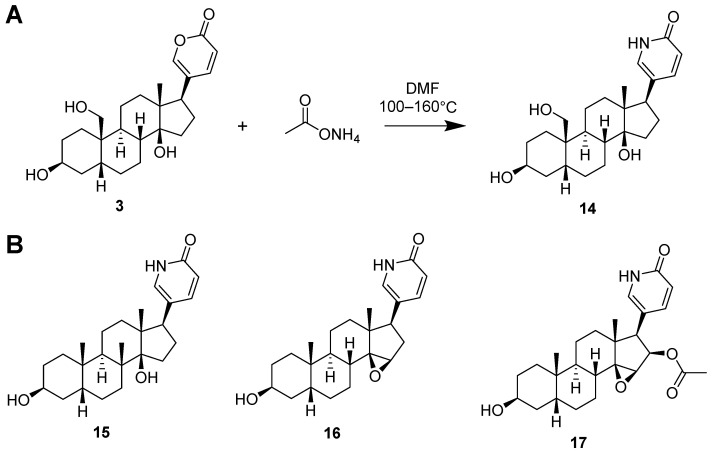
(**A**) The synthetic method of transforming lactone (**3**) into lactam (**14**). (**B**) Other three synthesized lactam derivatives, bufalin-lactam (**15**), resibufogenin-lactam (**16**), and cinobufagin-lactam (**17**) from compounds **5**, **7**, and **13**.

**Figure 6 molecules-29-01571-f006:**
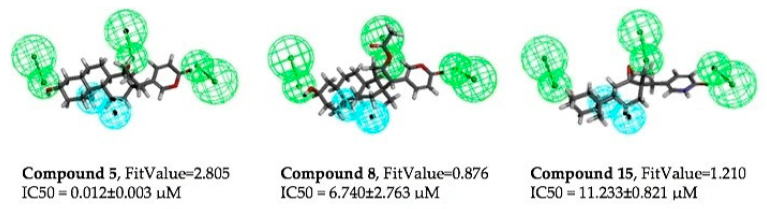
Common feature pharmacophore models. The IC_50_ values and FitValues for Bufalin (**5**), bufotalin (**8**), and bufalin-lactam (**15**) are presented. The green represents the hydrogen bond acceptors, and the blue represents the hydrophobic centers.

**Table 1 molecules-29-01571-t001:** ^1^H NMR (400 MHz) and ^13^C NMR (100 MHz) data for compounds **1** and **2** in CD_3_OD ^a^.

Position	1			2
	*δ* _C_	*δ*_H_ (*J* in Hz)	*δ* _C_	*δ*_H_ (*J* in Hz)
1	24.9, CH_2_	1.82 ^a^, 1.42 ^a^	24.7, CH_2_	1.80 ^a^, 1.46 ^a^
2	25.7, CH_2_	1.68 ^a^	27.5, CH_2_	1.82 ^a^, 1.26 ^a^
3	72.1, CH	5.05, br s	67.1, CH	4.06, (br. s)
4	31.4, CH_2_	2.00 ^a^, 1.48 ^a^	34.0, CH_2_	1.98 ^a^, 1.39 ^a^
5	30.3, CH	2.16 ^a^	30.7, CH	2.16 ^a^
6	22.1, CH_2_	1.84 ^a^, 1.34 ^a^	22.1, CH_2_	1.85 ^a^, 1.36 ^a^
7	27.2, CH_2_	1.87 ^a^, 1.24 ^a^	28.1, CH_2_	1.65 ^a^, 1.58 ^a^
8	42.8, CH	1.68 ^a^	43.0, CH	1.68 ^a^
9	36.6, CH	1.80 ^a^	36.4, CH	1.87 ^a^
10	40.4, C	-	39.6, C	-
11	22.6, CH_2_	1.46 ^a^, 1.22 ^a^	22.8, CH_2_	1.53 ^a^, 1.22 ^a^
12	42.2, CH_2_	1.50 ^a^, 1.42 ^a^	42.0, CH_2_	1.53 ^a^, 1.44 ^a^
13	49.8, C	-	49.7, C	-
14	86.2, C	-	86.1, C	-
15	33.0, CH_2_	2.16 ^a^, 1.73 ^a^	33.0, CH_2_	2.14 ^a^, 1.73 ^a^
16	29.8, CH_2_	2.18 ^a^, 1.73 ^a^	29.8, CH_2_	2.18 ^a^, 1.74 ^a^
17	52.3, CH	2.56, m	52.2, CH	2.56, dd (9.4, 6.0)
18	17.3, CH_3_	0.71, s	17.3, CH_3_	0.72, s
19	65.4, CH_2_	3.89 (d, 11.2); 3.38 (d, 11.2)	68.9, CH_2_	4.38 (d, 11.2); 4.00 (d, 11.2)
20	125.0, C	-	125.0, C	-
21	150.5, CH	7.43 (dd, 2.6, 1.0)	150.4, CH	7.44 (dd, 2.6, 1.0)
22	149.4, CH	7.99 (dd, 9.7, 2.6)	149.3, CH	8.00 (dd, 9.7, 2.6)
23	115.4, CH	6.28 (dd, 9.7, 1.0)	115.4, CH	6.29 (dd, 2.6, 1.0)
24	164.8, C	-	164.7, C	-
25	172.7, C	-	173.3, C	-
26	21.3, CH_3_	2.04, s	20.9, CH_3_	2.07, s

^a^ Overlapped signals are reported without designating multiplicity.

**Table 2 molecules-29-01571-t002:** The cytotoxic activity against prostate cancer cells of compounds **1**–**17** ^a^.

Compound	Cancer Cells (μM)	
	PC-3	DU145
Paclitaxel	0.031 ± 0.030	0.001 ± 0.001
**1**	0.162 ± 0.156	0.050 ± 0.015
**2**	0.214 ± 0.286	0.211 ± 0.221
**3**	0.014 ± 0.001	0.010 ± 0.007
**4**	0.091 ± 0.062	0.032 ± 0.010
**5**	0.012 ± 0.003	0.017 ± 0.014
**6**	0.040 ± 0.046	0.020 ± 0.016
**7**	0.259 ± 0.007	0.104 ± 0.054
**8**	6.740 ± 2.763	14.947 ± 5.074
**9**	0.571 ± 0.130	2.485 ± 0.772
**10**	0.035 ± 0.022	0.137 ± 0.131
**11**	0.040 ± 0.026	0.037 ± 0.003
**12**	0.007 ± 0.002	0.017 ± 0.007
**13**	0.050 ± 0.024	0.028 ± 0.006
**14**	6.261 ± 3.901	6.667 ± 2.744
**15**	11.233 ± 0.821	13.263 ± 1.506
**16**	>25	>25
**17**	56.297 ± 7.763	68.563 ± 14.742

Data were presented as mean ± S.E. The experiments were performed at least three times. ^a^ Paclitaxel was used as positive control.

## Data Availability

The original contributions presented in the study are included in the article/Appendix A. Further inquiries can be directed to the corresponding authors.

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
