# Peer review of "Isolation, Identification and Chemical Modification of Bufadienolides from Bufo melanostictus Schneider and Their Cytotoxic Activities against Prostate Cancer Cells"

_molecules, 2024, doi:10.3390/molecules29071571_

Round 1

Reviewer 1 Report

Comments and Suggestions for Authors

The two new bufadienolides (1 and 2), along with eleven known compounds (3-13) were isolated from Bufo melanostictus Schneider. Their structures and cytotoxic activities are well determined. I have no issues to suggest. 

The present paper aimed to isolate, identify and modify bufadienolides from Bufo melanostictus Schneider and to check their activity against prostate cancer.

This study contributes with preclinical evidence substantiating the therapeutic viability of bufadienolides and toad venom as intervention strategies for prostate cancer.

There are two new molecules well characterized by the appropriate spectroscopic and chromatographic methods.

The methodology used is well described and the results obtained are explained so I have no corrections.

The conclusions are consistent with the results and the discussion given. All main questions addressed were answered. According to the SAR the lactone ring of BDs is important in terms of antitumor  activity since the lactam derivatives of BDs lost their activities so this study provided new potential anti-tumor molecules for clinical treatment.

The references cited are appropriate and the majority are published in the last decade.

The quality of the tables and figures is satisfying and I have no corrections.  

Author Response

1.Summary

Thank you very much for taking the time to review this manuscript. Please find the detailed responses below and the corresponding revisions highlighted by using yellow colored text in the re-submitted files.

2.Point-by-point response to Comments and Suggestions for Authors

The two new bufadienolides (1 and 2), along with eleven known compounds (3-13) were isolated from Bufo melanostictus Schneider. Their structures and cytotoxic activities are well determined. I have no issues to suggest.

The present paper aimed to isolate, identify and modify bufadienolides from Bufo melanostictus Schneider and to check their activity against prostate cancer.

This study contributes with preclinical evidence substantiating the therapeutic viability of bufadienolides and toad venom as intervention strategies for prostate cancer.

There are two new molecules well characterized by the appropriate spectroscopic and chromatographic methods.

The methodology used is well described and the results obtained are explained so I have no corrections.

The conclusions are consistent with the results and the discussion given. All main questions addressed were answered. According to the SAR the lactone ring of BDs is important in terms of antitumor  activity since the lactam derivatives of BDs lost their activities so this study provided new potential anti-tumor molecules for clinical treatment.

The references cited are appropriate and the majority are published in the last decade.

The quality of the tables and figures is satisfying and I have no corrections.

Response: Thank you for your comment.

Reviewer 2 Report

Comments and Suggestions for Authors

13 Bufadienolides were extracted from Chinese medicine toad venom, including 2 new compounds that were characterized thoroughly through intensive NMR studies as well as X-ray diffraction. The cytotoxicity of all compounds was evaluated against two prostate cancer cell lines.

Since data for non-malignant cell lines is not available, it is recommended to include at least any relevant information about toxicity.

The mentioned previous studies demonstrating activity against prostate cancer cells should be cited in line 50.

Author Response

1.Summary

Thank you very much for taking the time to review this manuscript. Please find the detailed responses below and the corresponding revisions highlighted by using yellow colored text in the re-submitted files.

2.Point-by-point response to Comments and Suggestions for Authors

Comments 1: 13 Bufadienolides were extracted from Chinese medicine toad venom, including 2 new compounds that were characterized thoroughly through intensive NMR studies as well as X-ray diffraction. The cytotoxicity of all compounds was evaluated against two prostate cancer cell lines.

Since data for non-malignant cell lines is not available, it is recommended to include at least any relevant information about toxicity.

Response 1: Thank you for your suggestion. We have used the human nonmalignant prostate epithelial RWPE-1 cell line for parallel comparison. All compounds showed similar toxicity in RWPE-1 cells compared to the two prostate cancer cell lines. However, when we used African green monkey kidney-derived Vero cell line, the compounds showed much lower toxicity than in prostate cancer cell lines. The toxicity varies among different cell types. We have included this relevant information about toxicity in the discussion section.

Comments 2: The mentioned previous studies demonstrating activity against prostate cancer cells should be cited in line 50.

Response 2: Thank you for your suggestion. We have revised this section and added the references.

Reviewer 3 Report

Comments and Suggestions for Authors

Dear Authors,

Most of all, please follow "Instructions for Authors" of the journal.
I believe that an extensive revision of the manuscript should be required.
Especially, "References: References must be numbered in order of appearance in the text (including table captions and figure legends) and listed individually at the end of the manuscript".

[Major concerns]
1. There was no negative controls nor control cells. How could the authors conclude that the activities were specific against prostate cancer cells. The Authors should include negative controls in the study. The negative controls are not the cells without compounds.

[Minor concerns]
1. Figure 1 was not mentioned or explained in the text.

2. No figure legends and explanations for Fig.1, 2, 3, 4 and 6.

3. Fig. 2 appeared after Figs. 3 and 4 in the text.

4. (lines 101-106 and 115-119) These should be results.

5. (lines 94, 97 and 99) What were "method and water"?

6. (lines 94, 97, 98 and 99) What were "pre-HPLC" and "re-HPLC"?

7. (lines 137-138) The Authors should remove ", and their morphological changes were monitored microscopically".

8. (line 151) What was "Control group" here?

9. How did the Authors obtained COSY, HMBC correlations and NOESY correlations? These should be explained in "Materials and Methods" section.

10. The Authors should discuss why all four modified compounds (14-17) lost the activity.

Comments on the Quality of English Language

It was quite difficult to read the text. Please revise carefully.

Author Response

1.Summary

Thank you very much for taking the time to review this manuscript. Please find the detailed responses below and the corresponding revisions highlighted by using yellow colored text in the re-submitted files.

2.Point-by-point response to Comments and Suggestions for Authors

Most of all, please follow "Instructions for Authors" of the journal.

I believe that an extensive revision of the manuscript should be required.

Especially, "References: References must be numbered in order of appearance in the text (including table captions and figure legends) and listed individually at the end of the manuscript".

Response: Thank you for your suggestion. We have numbered the references both in the text and in the References section.

[Major concerns]

  1. There was no negative controls nor control cells. How could the authors conclude that the activities were specific against prostate cancer cells. The Authors should include negative controls in the study. The negative controls are not the cells without compounds.

Response: Thank you for your suggestion. We have added this section to the manuscript: "The cells treated with the vehicle only (0.1% DMSO in medium) served as the negative control, consistent with the solvent used for dissolving the sample. The absorbance of cells treated with medium containing 0.1% DMSO was regarded as 100%.” We also added a description of using these compounds in normal cells, although they showed similar toxic effects as compared to the data in prostate cancer cells. Clearly, more studies are needed before using these class of compounds in clinical settings.

[Minor concerns]

  1. Figure 1 was not mentioned or explained in the text.

Response: We have mentioned Figure 1 in the text.

  1. No figure legends and explanations for Fig.1, 2, 3, 4 and 6.

Response: We have added figure legends and explanations for Fig.1, 2, 3, 4 and 6.

  1. Fig. 2 appeared after Figs. 3 and 4 in the text.

Response: We have made the corresponding modifications.

  1. (lines 101-106 and 115-119) These should be results.

Response: We have made the corresponding modifications.

  1. (lines 94, 97 and 99) What were "method and water"?

Response: We have made the corresponding modifications.

  1. (lines 94, 97, 98 and 99) What were "pre-HPLC" and "re-HPLC"?

Response: We have made the corresponding modifications.

  1. (lines 137-138) The Authors should remove ", and their morphological changes were monitored microscopically".

Response: We have made the corresponding modifications.

  1. (line 151) What was "Control group" here?

Response: A pharmacophore represents the spatial arrangement of generic molecular interactions that succinctly elucidate the biological activity of a ligand molecule. Consequently, the construction of our pharmacophore model integrated structural information and bioactivity data of all compounds listed in our manuscript. Instead of categorizing chemicals into "Control" and "Treatment" groups, the ligands were divided into three groups based on their binding affinity using the 'Common Feature Pharmacophore' application within Discovery Studio. The first group, exhibiting higher binding affinity, was labeled as 2, while compounds with moderate or lower binding affinity were labeled as 1 and 0, respectively.

  1. How did the Authors obtained COSY, HMBC correlations and NOESY correlations? These should be explained in "Materials and Methods" section.

Response: We have made the corresponding modifications both in the "Materials and Methods" and “Results and Discussion” section.

  1. The Authors should discuss why all four modified compounds (14-17) lost the activity.

Response: We have made the corresponding modifications. “Structurally, changes in the cyclopentane conformation affect the size of the molecule, thereby influencing the overlap with hydrophobic and hydrogen bond feature elements, a revelation corroborated by the insights gleaned from Figure 6. Compared to bufalin (5), bufotalin (8) (16β-acetoxyl) exhibited decreased activity, which contrasts with previous reports [14] and may be attributed to the variety of cancer cells used in the study. The transformation of the bufadienolide structure (5) into an internal amide (15) resulted in a significant decrease in activity. The disruption of the lactone structure led to a decrease in activity, highlighting the crucial role of the lactone structure for the activity of these compounds.”

3.Response to Comments on the Quality of English Language

It was quite difficult to read the text. Please revise carefully.

Response: Thank you for your suggestion. We have thoroughly revised the entire manuscript and improved the language accordingly.

Round 2

Reviewer 3 Report

Comments and Suggestions for Authors

Dear Authors,

I would like to appreciate your effort to respond to all my concerns.
Now, I will highly recommend the manuscript to be accepted in the current form.

Best regards,
Kentaro Kato